# Quantitative Coronary CT Angiography and Pericoronary Adipose Tissue in Acute Myocardial Infarction: Relationship with Dynamic Myocardial Perfusion SPECT

**DOI:** 10.3390/diagnostics15222840

**Published:** 2025-11-09

**Authors:** Ayana Dasheeva, Darya Vorobeva, Kristina Kopeva, Alina Maltseva, Andrew Mochula, Irina Vorozhtsova, Elena Grakova, Konstantin Zavadovsky

**Affiliations:** 1Department of Radiology and Tomography, Cardiology Research Institute, Tomsk National Research Medical Center, Russian Academy of Sciences, Tomsk 634012, Russiakonstzav@gmail.com (K.Z.); 2Emergency Cardiology Department, Cardiology Research Institute, Tomsk National Research Medical Center, Russian Academy of Sciences, Tomsk 634012, Russia; darya.lipnyagova@yandex.ru; 3Department of Myocardial Pathology, Cardiology Research Institute, Tomsk National Research Medical Center, Russian Academy of Sciences, Tomsk 634012, Russia; 4Nuclear Department, Cardiology Research Institute, Tomsk National Research Medical Center, Russian Academy of Sciences, Tomsk 634012, Russia; 5Education Department, Cardiology Research Institute, Tomsk National Research Medical Center, Russian Academy of Sciences, Tomsk 634012, Russia

**Keywords:** acute myocardial infarction, MINOCA, MICAD, pericoronary adipose tissue, plaque burden, quantitative CCTA, dynamic SPECT

## Abstract

**Background/Objectives:** Despite growing evidence on quantitative computed tomography (CT) analysis of coronary plaques and pericoronary adipose tissue (PCAT), their association with myocardial perfusion (MP) in patients with first acute myocardial infarction (AMI) with obstructive coronary artery disease (MICAD) and non-obstructive coronary arteries (MINOCA) remain unclear. The aim of this study was to assess the relationship between quantitative CT coronary plaque components and PCAT characteristics with MP, myocardial blood flow (MBF) and coronary flow reserve (CFR) obtained by dynamic single-photon emission computed tomography (SPECT) in patients with AMI. **Methods:** Patients with a first episode of AMI were included in the study. All patients underwent coronary CT angiography with quantitative assessment of plaque volume (PV) and burden (PB), as well as PCAT volume and attenuation. Dynamic SPECT was performed on cadmium–zinc–telluride gamma-camera for quantitative assessment of MP parameters, stress and rest MBF, and CFR. **Results:** A total of 31 patients (median age 62 [56–70] years) were analyzed, including MICAD (*n* = 21) and MINOCA (*n* = 10). MICAD patients had significantly higher total PV and PB, mainly due to non-calcified and fibrofatty components (*p* < 0.05), while low-attenuation (LAP) and calcified plaques (CP) did not differ between groups. PCAT volumes were higher in MICAD (*p* < 0.05), whereas PCAT attenuation showed no differences. Dynamic SPECT revealed lower stress MBF and CFR in MICAD (*p* < 0.05). Correlation analysis showed positive associations of PV and PB with MP summed stress and rest scores, except LAP or CP; PB was negatively associated with MBF. In addition, PCAT volume correlated negatively with stress and rest MBF and CFR, as well as PCAT attenuation correlated positively with stress-induced MP abnormalities. **Conclusions**: Patients with MICAD demonstrated a greater extent of atherosclerosis and larger PCAT volume compared with MINOCA. Moreover, PCAT volume demonstrated inverse associations with MBF and CFR, indicating a potential link between PCAT characteristics and microvascular dysfunction.

## 1. Introduction

Acute myocardial infarction (AMI) remains a leading cause of mortality worldwide [1,2]. Most AMI are caused by obstructive coronary artery disease (MICAD); however, an estimated 5–15% present as myocardial infarction (MI) with non-obstructive coronary arteries (MINOCA) [3]. MINOCA is increasingly recognized as a distinct clinical entity, characterized by heterogeneous mechanisms, including coronary microvascular dysfunction, vasospasm, non-obstructive plaque rupture and thromboembolic events [4]. Differentiating MINOCA from MICAD is crucial for accurate diagnosis, risk stratification, and targeted therapeutic strategies.

Coronary CT angiography (CCTA) is established as the first-line non-invasive anatomical test for evaluating stable coronary artery disease (CAD), providing detailed 3 D coronary artery imaging with high diagnostic accuracy for stenosis detection (sensitivity 97%, specificity 78%) [5,6]. CCTA enables quantitative assessment, including plaque volume (PV) and plaque burden (PB), and plaque composition (low-attenuation, fibrofatty and calcified) [7]. Previous studies, including the ISCHEMIA trial (3711 patients, mean age 64 years, 79% male), demonstrated the prognostic significance of quantitative CCTA-derived plaque characteristics with atherosclerotic burden independently associated with cardiovascular mortality and AMI in patients with stable CAD [8]. At the same time, no data are available on differences in atherosclerotic burden derived from quantitative plaque analysis between MINOCA and MICAD patients.

Beyond the coronary artery, there is growing interest in pericoronary adipose tissue (PCAT), which is regarded as a metabolically active paracrine organ characterizing the inflammation [9]. CT-derived PCAT characteristics include volumetric and attenuation measures. While PCAT volume has been linked to systemic cardiometabolic risk and overall coronary plaque burden [10], PCAT attenuation, also known as fat attenuation index (FAI), has been proposed as a marker of perivascular inflammation and predictor of major adverse cardiovascular events (MACE) [11]. However, the role of these PCAT characteristics in AMI patients, specifically MICAD and MINOCA has not been determined.

Despite robust evidence supporting the prognostic value of PB, PV, and PCAT characteristics in stable CAD, there remains a clear knowledge gap in understanding their role and clinical implications following AMI. To our knowledge, no previous study has integrated quantitative CCTA and PCAT analysis with dynamic single-photon emission computed tomography (SPECT)-derived myocardial perfusion (MP), myocardial blood flow (MBF), and coronary flow reserve (CFR) in first AMI patients. Integrating these anatomical and functional parameters may provide complementary insights for improved risk stratification and inform personalized therapeutic decision-making in the post-infarction population.

Therefore, the aim of this study was to assess the relationship between quantitative CCTA plaque and PCAT characteristics with MP abnormalities, MBF and CFR obtained by dynamic SPECT in patients with AMI, including both obstructive and non-obstructive phenotypes.

## 2. Materials and Methods

### 2.1. Study Design

This retrospective, single-center cohort study was conducted using clinical and imaging data previously collected within a prospective, registered clinical trial (ClinicalTrials.gov Identifier: NCT03572023, registered on 28 June 2018). The original prospective study received approval from the Local Ethics Committee (Protocol No. 164, 23 November 2017). Written informed consent was obtained from all participants for study participation and secondary data analysis. Additional ethical approval for the retrospective evaluation was granted under Protocol No. 258, 10 January 2024.

The study was performed in accordance with the Declaration of Helsinki and Good Clinical Practice guidelines. Clinical and imaging data were acquired using the facilities and equipment of the Medical Genomics Centre at the Tomsk National Research Medical Centre (Tomsk, Russia).

### 2.2. Study Population

From 2017 to 2018 patients with first AMI were enrolled in the study. According to the presence of ≥50% coronary artery stenosis two groups were identified—MINOCA and MICAD. All patients received standard therapy for AMI according to national guidelines, including dual antiplatelet and lipid-lowering therapy, angiotensin-converting enzyme inhibitors, and beta-blockers in the absence of absolute contraindications.

Inclusion criteria: patients ≥18 years with first AMI, invasive coronary angiography (ICA) within 24 h of symptom onset, non-obstructive CAD (<50%) for MINOCA, obstructive (≥50%) stenosis in ≥1 coronary artery for MICAD, GRACE moderate/high risk, and sinus rhythm by electrocardiogram (ECG).

Exclusion criteria: hemodynamic instability, moderate/severe valvular disease, atrial fibrillation, severe comorbidities (advanced heart failure, ≥3b renal disease (eGFR < 45 mL/min/1.73 m^2^), chronic obstructive pulmonary disease stage III–IV, active malignancy, severe hepatic dysfunction), MI related to prior revascularization, acute myocarditis, Takotsubo cardiomyopathy, pacemaker, claustrophobia, contraindications to adenosine administration.

All patients underwent CCTA as well as dynamic SPECT by cadmium–zinc–telluride (CZT) gamma-camera.

Patient flow is shown in Figure 1.

### 2.3. Coronary CT Angiography Acquisition Protocol

CCTA was performed using a 64-slice CT scanner (Discovery NM/CT 570c, GE Healthcare, Milwaukee, WI, USA) on day 7 ± 2 following ICA (detailed time interval in Appendix A). Preparation included oral or intravenous beta-blockers to reduce heart rate and, when clinically indicated, oral prednisolone. Patients were instructed to avoid caffeine-containing food and beverages on the day of the examination and to discontinue medications such as metformin, sildenafil, and nonsteroidal anti-inflammatory drugs. Prior to scanning, all participants were screened for contraindications including iodinated contrast allergy, pregnancy, and renal impairment. Heart rate and blood pressure were measured immediately before image acquisition. All patients received 0.5 mg of sublingual nitroglycerin approximately 3–5 min prior to the scan to achieve coronary vasodilation.

For contrast-enhanced acquisition, 80–90 mL of nonionic iodinated contrast agent (Ultravist 370 mg/mL; Bayer, Leverkusen, Germany) was injected through an 18-gauge antecubital vein catheter at a rate of 5.0–5.5 mL/s, followed by a 50 mL saline flush using a dual-head injector (XD 8000 CT Motion; Ulrich Medical, Ulm, Germany).

The selection of ECG gating mode was based on heart rate: prospective ECG-triggered acquisition was applied in patients with heart rate < 65 beats per minute (bpm), and retrospective ECG gating was used in those with heart rate ≥ 65 bpm or in cases with arrhythmia. Acquisition parameters included a tube voltage of 120 kV and tube current modulation based on body mass index (range: 450–700 mA), with ECG-based dose modulation (maximum tube current during 40–80% of the R–R interval). The tube rotation time was 0.35 s, and the helical pitch was automatically adjusted according to heart rate (0.18:1–0.24:1).

The scan range extended from the level of the carina to the diaphragm to ensure complete coverage of the heart. Images were reconstructed at 75% of the R–R interval using a slice thickness and interval of 0.625 mm. In cases of motion artifacts, additional reconstructions were performed at other phases of the cardiac cycle (from 10% to 90%).

Post-processing was performed on a dedicated workstation (Advantage Workstation Version 4.6 or VolumeShare 7; GE Healthcare, Milwaukee, WI, USA) using the CardIQ Xpress 2.0 Reveal software. Image analysis included axial slices, curved multiplanar reformations, cross-sectional views, and thin-slab maximum intensity projections. Coronary artery segmentation was conducted according to the 16-segment model [12].

### 2.4. Quantitative CT Analysis of Coronary Plaques

Quantitative coronary plaques analysis was performed using the PlaqID tool (CardIQ Xpress 2.0 Reveal) within the CardIQ Xpress 2.0 Reveal software (GE Healthcare, Milwaukee, WI, USA). For each coronary artery, the proximal analysis boundary was defined at the coronary ostium, and the distal boundary was set at the point where the vessel diameter decreased to less than 1.5 mm. The software automatically delineated the contours of the vessel lumen and outer vessel wall. All segmentations were reviewed and manually adjusted by an >10 years experienced cardiac CT radiologist (K.Z.) to ensure anatomical accuracy.

PV was calculated as the space enclosed between the lumen and outer vessel wall boundaries. By applying predefined attenuation thresholds in the PlaqID tool, the software enabled segmentation of the lumen and classification of plaque components according to their Hounsfield Unit (HU) values as follows: low-attenuation −30 to 30 HU; fibrofatty 31 to 350 HU; calcified >350 HU [13].

Quantitative parameters included PV: total (TPV), low-attenuation (PV-LA), fibrofatty (PV-FF), total non-calcified (TPV-NC), calculated as the sum of PV-LA and PV-FF, calcified (PV-C). PB was also calculated for each component and expressed as a percentage of the total vessel volume, defined as the sum of the lumen volume and corresponding PV: total (TPB), low-attenuation (PB-LA), fibrofatty (PB-FF), total non-calcified (TPB-NC), calcified (PB-C).

Comprehensive details on patient preparation, CCTA acquisition protocol, and image processing procedures have been previously described in our earlier publication [14].

### 2.5. Quantitative CT Analysis of Pericoronary Adipose Tissue

Quantitative analysis of PCAT was conducted using CardIQ Xpress 2.0 software (GE Healthcare, Milwaukee, WI, USA) following the standardized approach described by Antonopoulos et al. [9]. In accordance with this validated protocol, PCAT was measured along the proximal 40 mm segments of the right coronary artery (RCA), the left anterior descending artery (LAD) and the left circumflex artery (LCX).

The left main coronary artery (LMA) was excluded from the analysis. This decision was based on the short length of the LMA, its anatomical variability, and close proximity to the aortic wall, which makes it difficult to distinguish pericoronary from paraaortic adipose tissue. These anatomical characteristics may introduce partial-volume effects and reduce measurement accuracy and reproducibility. In accordance with the recommendations of Antonopoulos et al., exclusion of the LMA ensures methodological consistency with the original validated PCAT quantification protocol and helps maintain analytical precision [9].

PCAT was manually segmented using the Brush tool within a circular region of interest (ROI) with a diameter equal to the sum of three diameters of the analyzed arterial segment. Voxels within the attenuation range of −190 to −30 HU were classified as adipose tissue [9]. After applying these predefined thresholds, the software automatically calculated the mean attenuation (HU) and total PCAT voxel volume (cm^3^) for each ROI.

### 2.6. Dynamic Myocardial Perfusion SPECT

Myocardial perfusion scintigraphy was performed using a CZT SPECT/CT system (Discovery NM/CT 570c, GE Healthcare, Haifa, Israel), following a two-day rest–stress protocol performed on average 8 ± 2 days after ICA (detailed time interval in Appendix A). Patients were instructed to fast overnight and avoid caffeine, methylxanthines, and antianginal medications (nitrates, beta-blockers, calcium channel blockers) for at least 24 h prior to the examination.

Rest imaging was initiated after intravenous administration of 99mTc-Sestamibi at a dose of 9 MBq/kg, followed by a saline flush. Dynamic ECG-gated list-mode acquisition (610 s) began 10 s prior to radiotracer injection. Standard gated rest imaging was acquired 40 min post-injection. Pharmacologic stress was induced via intravenous adenosine infusion (160 µg/kg/min for 4 min), with administration of a second tracer dose (9 MBq/kg) at minute 2, followed by a second dynamic acquisition. Delayed gated stress imaging was performed 40 min after stress tracer injection. A low-dose CT scan was performed for attenuation correction.

Image reconstruction was performed with Myovation for Alcyone (GE Healthcare) using a penalized likelihood iterative algorithm and Butterworth filter. Reconstructed images were reformatted into standard cardiac planes. Perfusion analysis was conducted using Corridor 4DM software (Invia, Ann Arbor, MI, USA). MP was assessed using a 17-segment model and a 5-point scoring system. The summed stress score (SSS), summed rest score (SRS), and summed difference score (SDS) were calculated. Left ventricular functional parameters were derived from 16-frame ECG-gated reconstructions.

Dynamic image datasets were reframed into 20 temporal frames and processed using the 4DM Reserve application. Time–activity curves were generated for the left ventricular chamber and left ventricular myocardium A one-tissue compartment model was applied to estimate MBF, and the Renkin–Crone model was used to convert tracer uptake to absolute MBF values. CFR was calculated as the ratio of stress to rest MBF.

Comprehensive details on patient preparation, the dynamic SPECT acquisition protocol, and image processing procedures have been previously described in our earlier publication [15].

### 2.7. Statistical Analysis

All statistical analyses were performed using Statistica software, version 10.0 (StatSoft Inc., Tulsa, OK, USA). The Shapiro–Wilk test was used to assess the normality of data distribution. Continuous variables were reported as medians with interquartile ranges (Me [Q1; Q3]), while categorical variables were summarized as absolute frequencies and percentages.

Comparisons between continuous variables were performed using nonparametric tests, including the Mann–Whitney U test. Categorical variables were compared using Fisher’s exact test. Associations between continuous variables were evaluated using Spearman’s rank correlation. To control for potential type I error due to multiple testing, *p*-values obtained from the correlation analyses were adjusted using the Bonferroni correction. A two-sided *p* < 0.05 was considered statistically significant after adjustment.

## 3. Results

### 3.1. Patient Characteristics

A total of 31 patients with AMI were enrolled, including 10 with MINOCA and 21 with MICAD. The MICAD group had a higher proportion of male patients compared to MINOCA (66.7% vs. 50.0%, *p* = 0.03). No significant age difference was observed between the groups (median age: 62 years [56–68] vs. 68 years [57–79], *p* = 0.35). ST-segment elevation MI (STEMI) occurred more frequent in MICAD patients (*p* = 0.03), who were also more likely to receive thrombolytic therapy (*p* = 0.01) and percutaneous coronary intervention (PCI) (*p* < 0.001). Left ventricular (LV) ejection fraction was significantly lower in the MICAD group compared with MINOCA, accompanied by a higher LV end-diastolic volume, and a trend toward increased LV end-systolic volume. The distribution of baseline cardiovascular risk factors, including hypertension, diabetes, dyslipidemia and smoking, was similar between the groups. Detailed clinical characteristics are summarized in Table 1.

### 3.2. Quantitative CCTA Characteristics

Quantitative CCTA analysis data revealed significantly greater TPV, PV-FF, TPV-NC, TPB, PB-FF, TPB-NC in the MICAD group compared to MINOCA (*p* < 0.05 for all). There were no significant differences observed in the volumes of low-attenuation plaque (LAP) and calcified plaque.

The full quantitative CCTA data are presented in Table 2.

### 3.3. Quantitative PCAT Characteristics

PCAT quantitative assessment results as well as intergroup comparison are presented in Table 3.

Quantitative coronary CT analysis demonstrated that patients with MICAD had a significantly greater PCAT volume compared with those diagnosed with MINOCA. In contrast, PCAT attenuation did not show statistically significant differences between the two groups.

### 3.4. Dynamic SPECT Parameters

Dynamic SPECT demonstrated that MICAD patients exhibited more pronounced stress and rest MP abnormalities, as well as significantly reduced stress MBF and CFR compared to patients with MINOCA (*p* < 0.05 for all). Detailed SPECT results are summarized in Table 4.

### 3.5. Relationship of Quantitative CCTA Characteristics with Dynamic SPECT Parameters

The correlation analysis between quantitative CT plaque characteristics and myocardial perfusion parameters is presented in Figure 2.

The SSS and SRS appeared to show positive associations with TPV-NC and TPB-NC, mainly driven by the fibrofatty plaque components, whereas no clear correlations were observed with calcified plaque components.

In contrast, quantitative flow parameters tended to demonstrate inverse relationships with CT-derived plaque characteristics, with the most pronounced negative trend observed for rest MBF.

### 3.6. Relationship of Quantitative PCAT Characteristics with Dynamic SPECT Parameters

The correlation analysis suggested potential associations between quantitative PCAT characteristics and dynamic SPECT parameters (Figure 3). PCAT volume tended to show inverse relationships with stress MBF across coronary territories. In contrast, PCAT attenuation values showed weak positive correlations, which were mainly limited to stress-induced perfusion abnormalities in the LCX, while no consistent associations with MBF or CFR were observed.

Representative examples of quantitative CCTA analysis, PCAT evaluation, and dynamic SPECT parameters from patients in the MINOCA and MICAD cohorts are shown in Figure 4 and Figure 5.

## 4. Discussion

In this study, we evaluated the association of quantitative coronary CT angiography derived atherosclerotic plaque characteristics with PCAT characteristics and MP indices in first AMI patients with non-obstructive and obstructive CAD. The main results of the study are the following: (1) Although MICAD and MINOCA patients did not differ in age or major risk factors, quantitative CCTA appeared to indicate higher global atherosclerotic burden in MICAD, likely driven by a greater non-calcified plaque component. (2) PCAT volume tended to be higher in MICAD than in MINOCA, whereas PCAT attenuation showed no evident difference between the groups. (3) In the overall cohort, rest perfusion abnormalities as well as rest MBF seemed to be related to CCTA-derived coronary plaque components, whereas CFR did not. (4) PCAT volume showed a weak but statistically significant correlation with stress MBF, but no relationship with semi-quantitatively assessed MP indices.

Overall, our findings suggest a possible trend that quantitative CCTA and PCAT analysis may provide preliminary insights into residual ischemic risk in patients with a first AMI. To the best of our knowledge, this appears to be the first study to integrate quantitative CCTA and PCAT analysis with dynamic SPECT parameters in first AMI patients.

### 4.1. Clinical Significance of the Quantitative CCTA and the Association with Dynamic SPECT in Patients with a First AMI

A recent Society of cardiovascular computed tomography (SCCT) consensus statement emphasized the importance of reporting both total plaque burden and plaque composition in clinical practice [13]. In line with these recommendations, multiple studies have consistently demonstrated the prognostic value of quantitative CCTA-derived PB and PV in patients with stable CAD. Bell et al., in a meta-analysis of 38 studies, showed that CCTA-derived PV-LA and TPV were the parameters most frequently and independently associated with MACE in stable CAD [16]. Dundas et al. reported that TPV was associated with one-year adverse clinical events, providing incremental predictive value over luminal stenosis [17]. Similarly, Meah et al. demonstrated that PB-LA was a major predictor of one-year death or recurrent MI in patients with suspected acute coronary syndrome (ACS) [18].

Our study showed that patients with MICAD had a significantly greater TPV and TPB than those with MINOCA, mainly due to a larger non-calcified component. This aligns with Williams et al., who demonstrated that PB-LA by CCTA was a strong predictor of MI in stable CAD [19]. Our results extend these findings by demonstrating that within the non-calcified pool, the fibrofatty fraction contributed most to the difference between MICAD and MINOCA, whereas PV-C did not differ. This novel observation highlights the fibrofatty component as a key determinant of plaque vulnerability in the early post-AMI setting, underscoring its potential value for improving risk stratification beyond traditional measures. Importantly, our data show that these quantitative differences are evident even in a first-AMI cohort, suggesting that CCTA-derived plaque characteristics can reveal clinically relevant distinctions beyond angiographic stenosis severity. Therefore, quantitative CCTA improves risk assessment not only before, but after AMI by providing a more comprehensive view than stenosis alone. In the context of MINOCA, this is particularly important, as Winnberg et al. showed that CCTA can reveal subclinical atherosclerosis with vulnerable features in some patients, even in the absence of obstructive stenoses [20].

The association of MP abnormalities on SPECT with quantitative CCTA characteristics has been demonstrated in several studies. Diaz-Zamudio et al. identified a significant relationship between PB-LA in vessels with intermediate stenosis (30–69%) and myocardial ischemia in patients with suspected or established CAD [21]. Similarly, Liu et al. showed that quantitative CCTA parameters, including TPV and TPB as well as non-calcified and LAP components, were associated with the frequency of myocardial ischemia in patients with suspected CAD [22]. With regard to studies in patients with MI, particular attention has been paid to dynamic SPECT parameters. Mochula et al. demonstrated that global and regional flow indices, including MBF and CFR, reflect the severity and extent of structural myocardial changes in the acute phase of MI and are more closely associated with myocardial injury than conventional SPECT parameters [23,24].

Our findings are consistent with the above studies in demonstrating a direct relationship between the severity of coronary atherosclerosis and myocardial ischemia. At the same time, this study is the first to demonstrate associations between quantitative CCTA characteristics and abnormalities in MP and MBF at both stress and rest, as assessed by dynamic CZT SPECT, in AMI cohort. This aligns with Zhang et al., who demonstrated that stress MBF is a better prognostic factor for MACE [25]. Furthermore, we established that, in this population, the TNC-PB showed the strongest association with impaired resting MBF. This phenomenon has been described previously and may be considered a manifestation of microvascular dysfunction [23,26,27].

### 4.2. Clinical Significance of PCAT Analysis and the Association with Dynamic SPECT in Patients with a First AMI

Quantitative assessment of PCAT has been increasingly recognized as a valuable imaging marker of coronary inflammation, complementing plaque characterization and providing prognostic information beyond conventional CCTA [9,11]. Recent studies and reviews emphasize that attenuation-derived metrics such as the FAI or FAI-score may enable risk reclassification and longitudinal monitoring, particularly in patients without obstructive CAD [28,29]. Furthermore, evidence suggests that combining FAI with other imaging biomarkers such as the coronary artery calcium score (CACS) may enhance the detection of subclinical inflammation and improve prognostic stratification, as these parameters capture distinct yet complementary aspects of coronary pathology—PB-C and perivascular inflammatory activity [30].

However, the prognostic role of PCAT remains debated: while Chatterjee et al. [31] and van Rosendael et al. [32] reported no association between PCAT attenuation and long-term outcomes in patients with suspected CAD, more recent evidence supports its association with future ACS independent of traditional risk factors [33]. Importantly, Diau et al. highlighted that combining PCAT assessment with AI-driven risk algorithms can improve prediction of MINOCA and MACE in patients with non-obstructive CAD [34]. Furthermore, a comprehensive review of 17 studies by Tan et al. reported that elevated RCA PCAT attenuation was consistently associated with MACE, whereas findings for other vessels were less robust [35]. The assessment of PCAT volume in patients with CAD has been investigated in only a few studies; however, the available evidence demonstrates its potential clinical utility. Sun et al. found that periplaque PCAT volume was higher in calcified and mixed plaques compared with non-calcified plaques, whereas proximal and segmental PCAT volumes did not differ significantly among plaque types [36].

Our study expands this evidence by demonstrating that PCAT volume was significantly higher in MICAD patients compared with MINOCA, while PCAT attenuation did not differ between the groups. These observations suggest distinct biological roles of PCAT metrics: volume likely reflects chronic vascular remodeling and cumulative adipose tissue expansion, consistent with its association with obstructive lesions in patients with AMI reported by Balcer et al. [37], whereas attenuation is more sensitive to transient inflammation and may be particularly relevant in stable CAD, as highlighted by Tan et al. [38].

Also, PCAT represents an important link between coronary inflammation and microvascular dysfunction. Nomura et al. demonstrated that higher perivascular inflammation assessed by CCTA was independently associated with reduced CFR on positron emission tomography (PET), particularly in patients with low CACS or with non-obstructive CAD [39]. Chen et al. confirmed that FAI was elevated in patients without obstructive CAD who exhibited impaired CFR as measured by PET, highlighting its value for detecting microcirculatory dysfunction beyond epicardial stenosis [40]. Wen et al. demonstrated that PCAT attenuation, but not volume, was associated with the hemodynamic significance of coronary artery stenosis on invasive fractional flow reserve, and that incorporating PCAT attenuation into CCTA significantly improved the diagnostic accuracy for identifying ischemic stenoses in patients with suspected CAD [41]. In patients with ACS, Sugiyama et al. reported that layered plaques identified by optical coherence tomography at the culprit site correlated with increased PCAT attenuation and impaired global CFR after primary or urgent PCI [42].

In our cohort PCAT volume showed an inverse correlation with stress MBF in all coronary arteries. Therefore, an increase in perivascular fat volume may reflect chronic vascular remodeling and contributes to impaired perfusion. These results are consistent with aforementioned data indicating that FAI captures transient inflammatory changes, while PCAT volume measurements may be more stable indicators of long-term disease severity. Taken together, the complementary assessment of PCAT volume and attenuation may improve risk stratification after AMI by integrating markers of both chronic structural burden and acute inflammatory activity.

### 4.3. Multimodality Imaging Framework and Clinical Implications

The present findings should be viewed in the context of the current non-invasive multimodality imaging framework, which emphasizes the complementary integration of anatomical and functional pathways for the comprehensive assessment of CAD. Rather than being considered as alternative diagnostic strategies, anatomical imaging (e.g., CCTA and quantitative PCAT analysis) and functional imaging (e.g., dynamic SPECT, PET, or stress perfusion cardiovascular magnetic resonance) provide synergistic information that reflects both the structural and physiological components of coronary pathology.

Recent literature has highlighted that this integrated approach enhances diagnostic accuracy, supports individualized patient management, and contributes to a deeper understanding of disease mechanisms across the spectrum of coronary syndromes. As noted by Bergamaschi et al. in their review, combining CCTA-derived plaque and PCAT characteristics with functional parameters (MP, MBF and CFR) represents a pathophysiology-driven paradigm for evaluating CAD [43].

By applying this integrated framework in our cohort of AMI patients, we extend its application beyond chronic coronary syndrome and demonstrate that a combined anatomical–functional strategy may be particularly valuable in the post-infarction setting, where residual flow abnormalities and pericoronary inflammation often coexist. This perspective supports the evolving concept of multimodality imaging as a cornerstone of precision cardiology, enabling more accurate risk stratification and guiding personalized therapeutic decision-making.

This study has several limitations. First, the relatively small sample size may limit the generalizability of the findings, and further validation in larger cohorts is warranted. Second, despite the use of standardized imaging protocols and post-processing methods, some degree of measurement variability cannot be excluded. To reduce this limitation, we applied standardized threshold values for plaque components as recommended in the latest SCCT consensus documents. This approach increases the comparability of our results with those obtained using dedicated AI-based software. Third, medication use during hospitalization may have influenced both MBF and PCAT attenuation values and thus should be considered potential confounding factors when interpreting the results. Finally, the study did not include an economic assessment of the multimodality imaging approach, as this was beyond the scope and expertise of the research team; however, future studies could explore the cost-effectiveness of such strategies in clinical practice.

## 5. Conclusions

The present study provides preliminary evidence on the relationships between quantitative CCTA-derived plaque and PCAT characteristics and perfusion parameters measured by dynamic SPECT in patients with a first AMI.

Compared with MINOCA, the MICAD phenotype demonstrated a greater atherosclerotic burden, mainly associated with the fibrofatty plaque component, and a higher PCAT volume, suggesting that the extent of coronary atherosclerosis and PCAT expansion may contribute to the pathophysiological differences between these two conditions. Moreover, PCAT volume showed inverse correlations with MBF and CFR, indicating a potential link between PCAT and microvascular dysfunction.

These findings support the concept that a combined evaluation of coronary PB and PCAT characteristics may provide complementary insights into the pathophysiology of AMI with and without obstructive lesions. The integration of quantitative anatomical (CCTA) and functional (dynamic SPECT) imaging could enhance diagnostic accuracy, deepen understanding of CAD mechanisms, and support more individualized management strategies in this patient population.

## Figures and Tables

**Figure 1 diagnostics-15-02840-f001:**
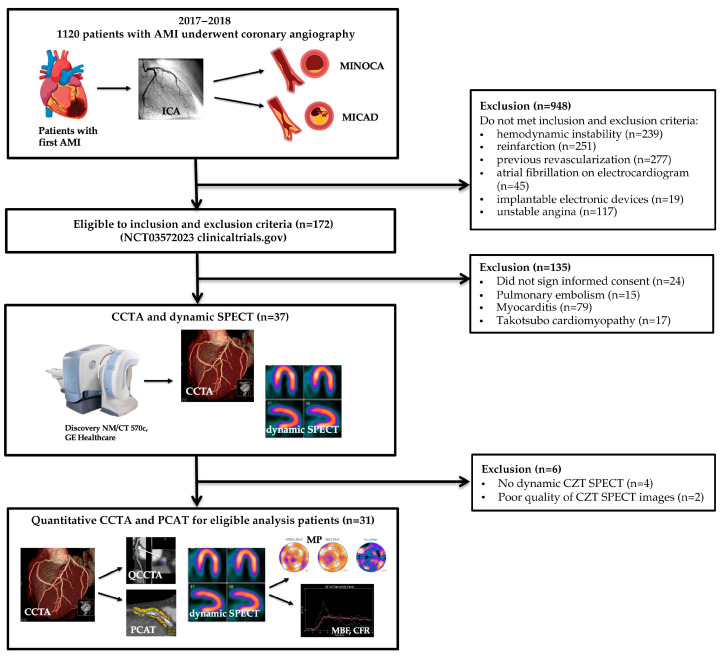
AMI—acute myocardial infarction, CCTA—coronary computed tomography angiography, CZT—cadmium-zinc-telluride, MINOCA—myocardial infarction with non-obstructive coronary artery disease, MICAD—myocardial infarction with obstructive coronary artery disease, SPECT—single-photon emission computed tomography, PCAT—pericoronary adipose tissue.

**Figure 2 diagnostics-15-02840-f002:**
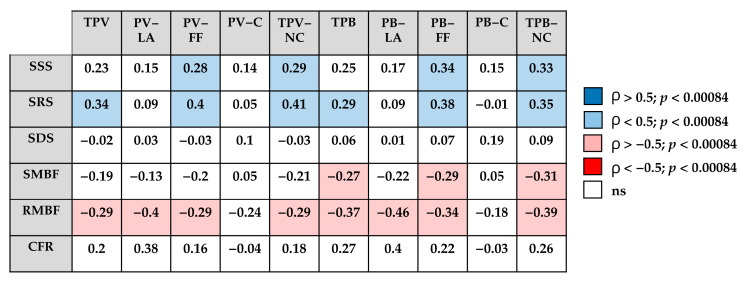
Correlation matrix between quantitative CCTA characteristics and dynamic SPECT parameters. TPV—total plaque volume; PV-LA—low-attenuation plaque volume; PV-FF—fibrofatty plaque volume; PV-C—calcified plaque volume; TPV-NC—non-calcified plaque volume; TPB—total plaque burden; PB-LD—low-attenuation plaque burden; PB-FF—fibrofatty plaque burden; PB-C—calcified plaque burden; TPB-NC—non-calcified plaque burden; SSS—summed stress score; SRS—summed rest score; SDS—summed difference score; SMBF—stress myocardial blood flow; RMBF—rest myocardial blood flow; CFR—coronary flow reserve; blue—positive correlation (ρ > 0.5, *p* < 0.00084); light blue—positive correlation (ρ < 0.5, *p* < 0.00084); pink—negative correlation (ρ < −0.5, *p* < 0.00084); red—strong negative correlation (ρ < −0.5, *p* < 0.00084); white—not significant.

**Figure 3 diagnostics-15-02840-f003:**
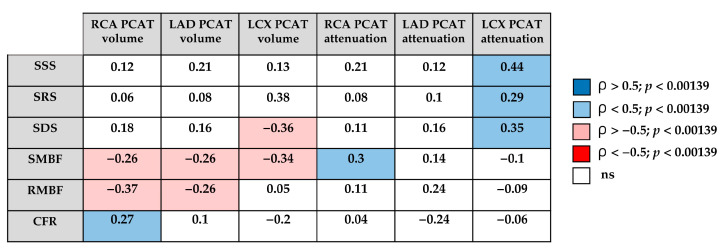
Correlation matrix between quantitative PCAT characteristics and dynamic SPECT parameters. RCA—right coronary artery; LAD—left anterior descending artery; LCX—left circumflex artery, PCAT—pericoronary adipose tissue; SSS—summed stress score; SRS—summed rest score; SDS—summed difference score; SMBF—stress myocardial blood flow; RMBF—rest myocardial blood flow; CFR—coronary flow reserve; blue—positive correlation (ρ > 0.5, *p* < 0.00139); light blue—positive correlation (ρ < 0.5, *p* < 0.00139); pink—negative correlation (ρ < −0.5, *p* < 0.00139); red—strong negative correlation (ρ < −0.5, *p* < 0.00139); white—not significant.

**Figure 4 diagnostics-15-02840-f004:**
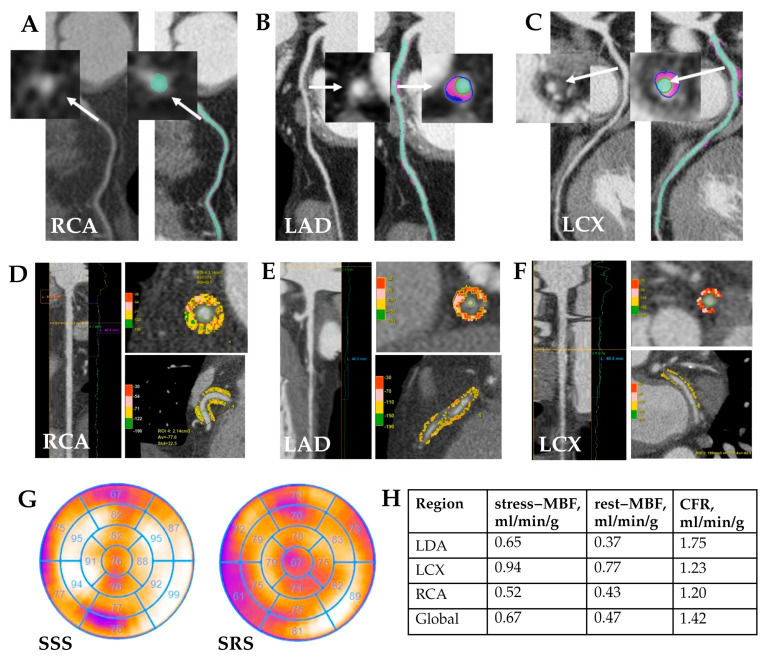
MINOCA patient clinical case study. Clinical case: Patient K., age 57, male. Quantitative CCTA results (**A**–**C**): coronary atherosclerosis with non-calcified and fibrofatty components in proximal LAD—stenosis 35%; in proximal LCX—stenosis 30%. White arrows indicate the atherosclerotic plaque before and after quantitative analysis. Quantitative PCAT (**D**–**F**): RCA—PCAT attenuation −77.6 HU, PCAT volume 2140 mm^3^; LAD—PCAT attenuation −72.1 HU, PCAT volume 823 mm^3^; LCX—PCAT attenuation −62.9 HU, PCAT volume 195 mm^3^. Myocardial SPECT (**G**) reviled abnormal myocardial perfusion at stress in the anterior and posterior walls (SSS 4); normal myocardial perfusion at rest. There is a mild to moderate impairment of myocardial blood flow reserve in all coronary arteries by dynamic SPECT (**H**). RCA—right coronary artery, LAD—left anterior descending coronary artery, LCX—left circumflex artery, MBF—myocardial blood flow, CFR—coronary flow reserve.

**Figure 5 diagnostics-15-02840-f005:**
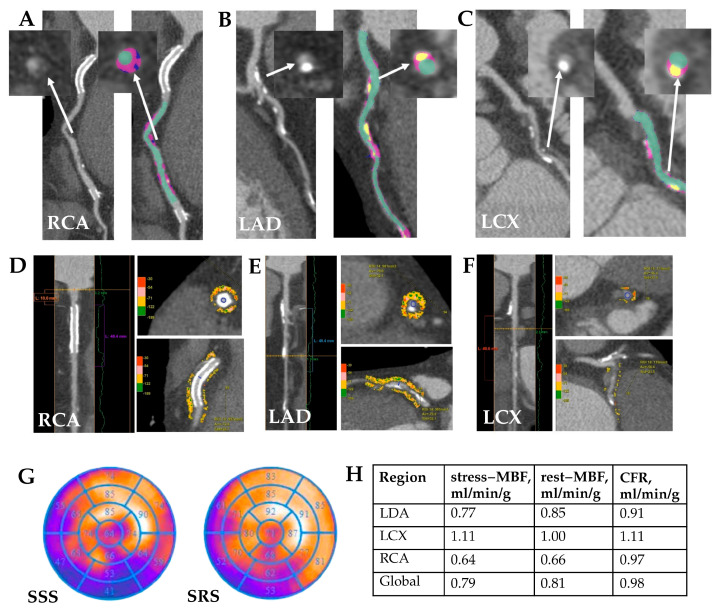
MICAD patient clinical case study. Clinical case: Patient K., age 69, male. Quantitative CCTA (**A**–**C**): A stent is visualized in the proximal RCA, non-calcified (low-attenuation and fibrofatty) and calcified components of atherosclerotic plaques in the proximal and mid RCA with a maximal stenosis of 65%; diffuse coronary atherosclerosis, with low-attenuation and calcified components of atherosclerotic plaques is visualized along the entire LAD with a maximal stenosis of 40%; low-attenuation and calcified components of atherosclerotic plaques in LCX artery with a maximal stenosis of 45%. White arrows indicate the atherosclerotic plaque before and after quantitative analysis. Quantitative PCAT (**D**–**F**): RCA—PCAT attenuation −72 HU, PCAT volume 997 mm^3^; LAD—PCAT attenuation −79 HU, PCAT volume 981 mm^3^; LCX—PCAT attenuation −56.4 HU, PCAT volume 119 mm^3^ Myocardial perfusion SPECT (**G**) reviled large stable perfusion defect in the inferior, inferior-lateral and inferior-septal wall and stress-induced perfusion defect in the inferior-lateral wall (SSS 17; SRS 9). There is a significant reduction in stress MBF and CFR in all coronary artery territories (**H**). RCA—right coronary artery, LAD– left descending coronary artery, LCX—left circumflex artery, MBF—myocardial blood flow, CFR—coronary flow reserve.

**Table 1 diagnostics-15-02840-t001:** Demographic and clinical characteristics of the study population.

Parameter	Whole Sample, *n* = 31	Group 1, *n* = 10 MINOCA	Group 2, *n* = 31 MICAD	*p*-Value
Male sex, *n* (%)	19 (61.3%)	5 (50.0%)	14 (66.6%)	0.03
Age, years	62 (56; 70)	68 (57; 79)	62 (56; 68)	0.35
Hypertension, *n* (%)	24 (77.4%)	10 (100%)	14 (66.6%)	0.61
Dyslipidemia, *n* (%)	25 (80.6%)	10 (100%)	15 (71.4%)	0.08
Obesity, *n* (%)	12 (38.7%)	3 (30.0%)	9 (42.8%)	0.13
Smoking, *n* (%)	13 (41.9%)	3 (30.0%)	10 (47.6%)	0.36
Type 2 diabetes mellitus, *n* (%)	4 (12.9%)	0 (0%)	4 (19.0%)	0.06
eGFR, mL/min/1.73 m^2^, Me (Q25; Q75)	72.0 (54.0; 89.0)	64.5 (53.0; 72.0)	77.0 (65.0; 92.0)	0.7
Stroke, *n* (%)	2 (6.5%)	0 (0%)	2 (9.5%)	0.27
STEMI, *n* (%)	23 (74.2%)	6 (60.0%)	17 (80.9%)	0.03
GRACE, %	2.0 (2.0; 5.0)	2.0 (2.0; 4.0)	2.2 (2.0; 5.0)	0.60
Thrombolysis in hospital, *n* (%)	12 (38.7%)	1 (10.0%)	11 (52.4%)	0.01
TIMI 2 flow, *n* (%)	6 (19.4%)	5 (50.0%)	1 (4.8%)	0.0075
PCI, *n* (%)	16 (51%)	0 (0%)	16 (76%)	0.0001
LVEF, %	60 (53; 65)	65 (64; 69)	57 (52; 63)	0.01
ESV, mL	42.5 (32; 53)	33 (25; 41)	45 (36; 58)	0.07
EDV, mL	103 (86; 119)	95 (80; 106)	105 (93; 123)	0.009
Length of hospital stay, days	11 ± 2	11 ± 2	11 ± 2	0.9

AMI—acute myocardial infarction; CAD—coronary artery disease; EDV—end-diastolic volume; eGFR—estimated glomerular filtration rate, ESV—end-systolic volume; GRACE—global registry of acute coronary events; LVEF—left ventricular ejection fraction; MINOCA—myocardial infarction with non-obstructive coronary arteries; MICAD—myocardial infarction with obstructive coronary artery disease; PCI—percutaneous coronary intervention; STEMI—myocardial infarction with ST-elevation; TIMI—thrombolysis in myocardial infarction; *p*-value—Mann–Whitney U test between MINOCA and MICAD groups, bold *p*-values indicate statistical significance (*p* < 0.05).

**Table 2 diagnostics-15-02840-t002:** Quantitative CCTA characteristics in MINOCA and MICAD groups.

Parameter	Whole Sample, *n* = 31	Group 1, *n* = 10 MINOCA	Group 2, *n* = 31 MICAD	*p*-Value
TPV, mm^3^	514.1 (349.8; 992.7)	408.8 (167; 756.3)	542.4 (355.2; 1070.2)	0.02
PV-LA, mm^3^	31.6 (17.5; 68.1)	31.6 (14.8; 41.6)	32.8 (18.8; 74.2)	0.29
PV-FF, mm^3^	416.6 (275.9; 877.7)	307.2 (143.9; 679.7)	459.1 (302.9; 928.6)	0.03
TPV-NC, mm^3^	439.3 (293; 918)	338.7 (164.5; 739.4)	479.9 (349.8; 1001)	0.02
PV-C, mm^3^	22.8 (10.9; 58.5)	16.9 (13; 48.6)	28.8 (10.9; 58.5)	0.43
TPB, %	21.8 (17.2; 33.8)	13.2 (10.8; 27)	23 (18.4; 37)	0.001
PB-LA, %	1.7 (1; 4)	1.2 (0.9; 1.9)	1.8 (1.1; 4.9)	0.11
PB-FF, %	18.3 (13.3; 29.2)	11.3 (6.8; 25.3)	18.7 (16.3; 31.6)	0.001
TPB-NC, %	19.8 (14; 30.9)	12 (10.7; 26.2)	19.9 (17.9; 33.3)	0.001
PB-C, %	1.5 (0.5; 3.3)	0.8 (0.6; 1.5)	1.9 (0.4; 3.7)	0.19

TPV—total plaque volume; PV-LA—low-attenuation plaque volume; PV-FF—fibrofatty plaque volume; PV-C—calcified plaque volume; TPV-NC—non-calcified plaque volume; TPB—total plaque burden; PB-LD—low-attenuation plaque burden; PB-FF—fibrofatty plaque burden; PB-C—calcified plaque burden; TPB-NC—non-calcified plaque burden; *p*-value—Mann–Whitney U test between MINOCA and MICAD groups, bold *p*-values indicate statistical significance (*p* < 0.05).

**Table 3 diagnostics-15-02840-t003:** Quantitative PCAT characteristics in MINOCA and MICAD groups.

Parameter	Whole Sample, *n* = 31	Group 1, *n* = 10 MINOCA	Group 2, *n* = 31 MICAD	*p*-Value
RCA PCAT volume, cm^3^	0.95 (0.6; 1.65)	0.58 (0.5; 1.2)	1.1 (0.71; 1.89)	0.01
LAD PCTA volume, cm^3^	0.74 (0.5; 1.62)	0.57 (0.48; 0.67)	1.27 (0.6; 1.79)	0.002
LCX PCAT volume, cm^3^	0.44 (0.31; 0.68)	0.35 (0.28; 0.44)	0.53 (0.32; 0.73)	0.04
RCA PCAT attenuation, HU	−72.7 (−78.1; −67.7)	−72.1 (−73.3; −67.7)	−73.7 (−81.8; −68.5)	0.2
LAD PCAT attenuation, HU	−70 (−75.8; −63.5)	−68.5 (−74.5; −61.1)	−70 (−77; −63.9)	0.3
LCX PCAT attenuation, HU	−64.25 (−70; −58.3)	−66.3 (−70.2; −61)	−64 (−68.7; −58)	0.21

RCA—right coronary artery; LAD—left anterior descending artery; LCX—left circumflex artery, PCAT—pericoronary adipose tissue, *p*-value—Mann–Whitney U test between MINOCA and MICAD groups, bold *p*-values indicate statistical significance (*p* < 0.05).

**Table 4 diagnostics-15-02840-t004:** Dynamic SPECT parameters in MINOCA and MICAD groups.

Parameter	Whole Sample, *n* = 31	Group 1, *n* = 10 MINOCA	Group 2, *n* = 31 MICAD	*p*-Value
SSS	6.5 (5; 12)	5 (4; 5)	9 (5; 13)	<0.001
SRS	4 (2; 7)	2 (1; 3)	6 (3; 11)	<0.001
SDS	3 (2; 5)	3 (2; 4)	4 (2; 5)	0.09
SMBF, mL/min/g	1.03 (0.89; 1.77)	2.02 (1.73; 2.18)	0.97 (0.8; 1.04)	<0.001
RMBF, mL/min/g	0.75 (0.54; 1.15)	0.88 (0.57; 1.24)	0.74 (0.45; 1.11)	0.27
CFR	1.33 (0.98; 1.99)	1.85 (1.33; 2.22)	1.23 (0.98; 1.86)	0.02

SSS—summed stress score; SRS—summed rest score; SDS—summed difference score; SMBF—stress myocardial blood flow; RMBF—rest myocardial blood flow; CFR—coronary flow reserve, *p*-value—Mann–Whitney U test between MINOCA and MICAD groups, bold *p*-values indicate statistical significance (*p* < 0.05).

## Data Availability

The data presented in this study are available on request from the corresponding author. The data are not publicly available due to privacy.

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
