# Peer review of "Quantitative Coronary CT Angiography and Pericoronary Adipose Tissue in Acute Myocardial Infarction: Relationship with Dynamic Myocardial Perfusion SPECT"

_diagnostics, 2025, doi:10.3390/diagnostics15222840_

Round 1
Reviewer 1 Report
Comments and Suggestions for Authors
This is an interesting manuscript that investigates the relationship between quantitative coronary CT angiography (CCTA) and pericoronary adipose tissue (PCAT) characteristics with myocardial perfusion (MP) and flow parameters in acute myocardial infarction (AMI) patients using dynamic SPECT. The integration of anatomical and functional imaging adds novelty and clinical relevance to the study. The manuscript follows a clear structure and is generally well-written. The study design is appropriate, and the methodology is detailed enough for reproducibility. However, some methodological clarifications are needed to improve transparency and data presentation.
Major Comments:
- The introduction clearly outlines the clinical context, but the specific hypothesis could be stated more explicitly. Consider rephrasing the last paragraph to emphasize what gap this study uniquely fills (e.g., “To our knowledge, no previous study has integrated quantitative CCTA and PCAT analysis with dynamic SPECT in first AMI patients.”).
- Please provide clearer definitions for exclusion criteria such as “myocardial inflammation” and “severe comorbidities”.
- Indicate the time interval between AMI onset and CCTA/SPECT acquisition (e.g., within 7 ± 2 days), since timing may influence PCAT and perfusion measurements.
- Clarify the reported GRACE score: a median of 2.0 seems inconsistent with typical GRACE scoring ranges; please verify or clarify the metric used.
- Statistical Analysis: indicate whether corrections for multiple testing were applied in the correlation analyses, given the number of variables examined.
- Figure 1 should appear in the Results section rather than in Methods.
- Tables 1 is informative but would benefit from including additional clinical data such as eGFR and medications at admission.
- The discussion could benefit from a broader contextualization within the current multimodality imaging framework, highlighting how the integration of anatomical and functional approaches recently emphasized in the literature (cite PMID: 37370978) supports a more comprehensive assessment of coronary disease across different clinical settings.
- The manuscript would benefit from a brief consideration of the economic implications of the proposed imaging approach. Including a short discussion on cost-effectiveness, perhaps referencing comparative analyses of alternative diagnostic strategies, would provide a more comprehensive perspective on its potential clinical applicability (cite PMID: 18510723).
- In the Discussion or Limitations section, please include a statement acknowledging potential confounders, particularly the possible effects of medications on MBF and PCAT attenuation.
Minor Comments:
- Abstract: move the number of enrolled patients (n = 31) from the Methods section to the Results, as per standard reporting practice.
- Abbreviations: Ensure consistency in the use of PCAT, PB, PV, and FAI across text and tables.
- Figures 4 and 5: The image captions are detailed but could be shortened; focus on the main imaging findings.
- References: The list is comprehensive and current. Ensure consistent reference formatting according to Diagnostics journal style.
Author Response
- Summary
We sincerely thank the reviewer for the evaluation of our manuscript, as well as for the constructive and insightful comments that helped improve the overall quality, clarity, and scientific rigor of the paper.
Detailed responses to each comment are provided in this document, and all changes have been marked using the comment function in the re-submitted version for easier review.
We greatly appreciate the time and effort invested by the reviewer and the editorial team in refining our work and are grateful for the opportunity to revise and resubmit our manuscript to Diagnostics.
- Point-by-point response to Comments and Suggestions for Authors
Major comments
Comments 1: The introduction clearly outlines the clinical context, but the specific hypothesis could be stated more explicitly. Consider rephrasing the last paragraph to emphasize what gap this study uniquely fills (e.g., “To our knowledge, no previous study has integrated quantitative CCTA and PCAT analysis with dynamic SPECT in first AMI patients.”).
Response 1: We thank the reviewer for this comment. We have revised the last paragraph of the Introduction to emphasize the study gap and hypothesis. The revised paragraph now highlights that, to our knowledge, no previous study has combined quantitative CCTA and PCAT analysis with dynamic SPECT in first AMI patients, thereby underscoring the novelty and rationale of our research.
The last paragraph has been updated in the manuscript on page 3, line 97, and now reads:
[Despite robust evidence supporting the prognostic value of PB, PV, and PCAT characteristics in stable CAD, there remains a clear knowledge gap in understanding their role and clinical implications following AMI. To our knowledge, no previous study has integrated quantitative CCTA and PCAT analysis with dynamic single-photon emission computed tomography (SPECT)–derived myocardial perfusion (MP), myocardial blood flow (MBF), and coronary flow reserve (CFR) in first AMI patients. Integrating these anatomical and functional parameters may provide complementary insights for improved risk stratification and inform personalized therapeutic decision-making in the post-infarction population.]
Comments 2: Please provide clearer definitions for exclusion criteria such as “myocardial inflammation” and “severe comorbidities”.
Response 2: We thank the reviewer for this helpful comment. Accordingly, we have revised the Methods section to provide explicit clarification.
The text has been updated in the manuscript on page 3, line 140, and now reads:
[Exclusion criteria: hemodynamic instability, moderate/severe valvular disease, atrial fibrillation, severe comorbidities (advanced heart failure, ≥3b renal disease (eGFR <45 ml/min/1,73m2), chronic obstructive pulmonary disease stage III–IV, active malignancy, severe hepatic dysfunction), MI related to prior revascularization, acute myocarditis, Takotsubo cardiomyopathy, pacemaker, claustrophobia, contraindications to adenosine administration].
Comments 3: Indicate the time interval between AMI onset and CCTA/SPECT acquisition (e.g., within 7 ± 2 days), since timing may influence PCAT and perfusion measurements.
Response 3: We thank the reviewer for this valuable suggestion. We have included a supplementary table (Table S1) presenting the exact time interval between AMI onset and imaging procedures for each individual patient.
The text has been updated in the manuscript on page 4, line 159, and now reads:
[CCTA was performed using a 64-slice CT scanner (Discovery NM/CT 570c, GE Healthcare, USA) on day 7 ± 2 following ICA (detailed time interval in Table S1)].
The text has been updated in the manuscript on page 6, line 240, and now reads:
[Myocardial perfusion scintigraphy was performed using a CZT SPECT/CT system (Discovery NM/CT 570c, GE Healthcare, Israel), following a two-day rest-stress protocol performed on average 8 ± 2 days after ICA (detailed time interval in Table S1)].
Comments 4: Clarify the reported GRACE score: a median of 2.0 seems inconsistent with typical GRACE scoring ranges; please verify or clarify the metric used.
Response 4: We thank the reviewer for this careful observation. We acknowledge that the initially reported GRACE score value could have caused confusion. The value of 2.0 refers not to the GRACE score itself but to the estimated in-hospital mortality risk (%) according to the GRACE risk model. The risk was calculated using an automatic calculator, where low risk corresponds to mortality less than 1% (score <109), moderate risk to mortality between 1–3% (score 109–140), and high risk to mortality above 3% (score >140).
In the revised version of the manuscript, we have also clarified this point in Table 1 by indicating the measurement unit as a percentage (%). The changes have been made in the Table 1 on page 7.
Comments 5: Statistical Analysis: indicate whether corrections for multiple testing were applied in the correlation analyses, given the number of variables examined.
Response 5: We thank the reviewer for this important comment. We have included a detailed explanation in the Statistical Analysis section, specifying that the Bonferroni correction was applied (Figure 2, Figure 3).
The text has been updated in the manuscript on page 5, line 277, and now reads:
To control for potential type I error due to multiple testing, p-values obtained from the correlation analyses were adjusted using the Bonferroni correction.
Comments 6: Figure 1 should appear in the Results section rather than in Methods.
Response 6: We sincerely thank the reviewer for this helpful suggestion. Figure 1 presents the overall study design and patient selection process, which we originally included in the Methods section to assist readers in understanding the study workflow. We believe that keeping it in this section helps maintain logical continuity between the description of the study population and subsequent analyses. However, we fully respect the reviewer’s opinion and, if he preferred, we will relocate Figure 1 to the Results section in the revised version.
Comments 7: Tables 1 is informative but would benefit from including additional clinical data such as eGFR and medications at admission.
Response 7: We thank the reviewer for this valuable comment. In accordance with the recommendation, we have added eGFR values to Table 1 to provide a more comprehensive clinical characterization of the study population. The changes have been made in the Table 1 on page 7.
Regarding medication therapy during hospitalization, this information has been clarified in the Methods section of the revised manuscript. Since the treatment was uniform across all patients, we preferred to describe it in the text rather than in the table.
The text has been updated in the manuscript on page 3, line 128, and now reads:
[Аll patients received standard therapy for AMI according to national guidelines, including dual antiplatelet and lipid-lowering therapy, angiotensin-converting enzyme inhibitors, and beta-blockers in the absence of absolute contraindications].
Comments 8: The discussion could benefit from a broader contextualization within the current multimodality imaging framework, highlighting how the integration of anatomical and functional approaches recently emphasized in the literature (cite PMID: 37370978) supports a more comprehensive assessment of coronary disease across different clinical settings.
Response 8: We thank the reviewer for this constructive and insightful suggestion. In the revised version of the manuscript, we have therefore included an additional subsection in the Discussion that specifically addresses this concept.
The new subsection is titled “Multimodality Imaging Framework and Clinical Implications” and has been added to the Discussion on page 15 and now reads:
[Multimodality Imaging Framework and Clinical Implications
The present findings should be viewed in the context of the current non-invasive multimodality imaging framework, which emphasizes the complementary integration of anatomical and functional pathways for the comprehensive assessment of CAD. Rather than being considered as alternative diagnostic strategies, anatomical imaging (e.g., CCTA and quantitative PCAT analysis) and functional imaging (e.g., dynamic SPECT, PET, or stress perfusion cardiovascular magnetic resonance) provide synergistic information that reflects both the structural and physiological components of coronary pathology.
Recent literature has highlighted that this integrated approach enhances diagnostic accuracy, supports individualized patient management, and contributes to a deeper understanding of disease mechanisms across the spectrum of coronary syndromes. As noted by Bergamaschi et al. in their review, combining CCTA-derived plaque and PCAT characteristics with functional parameters (MP, MBF, and CFR) represents a pathophysiology-driven paradigm for evaluating CAD [42].
By applying this integrated framework in our cohort of acute myocardial infarction patients, we extend its application beyond chronic coronary syndrome and demonstrate that a combined anatomical-functional strategy may be particularly valuable in the post-infarction setting, where residual flow abnormalities and pericoronary inflammation often coexist. This perspective supports the evolving concept of multimodality imaging as a cornerstone of precision cardiology, enabling more accurate risk stratification and guiding personalized therapeutic decision-making.]
Comments 9: The manuscript would benefit from a brief consideration of the economic implications of the proposed imaging approach. Including a short discussion on cost-effectiveness, perhaps referencing comparative analyses of alternative diagnostic strategies, would provide a more comprehensive perspective on its potential clinical applicability (cite PMID: 18510723).
Response 9: We sincerely thank the reviewer for this thoughtful and constructive suggestion. At the same time, the primary aim of our study was to analyze the relationships between quantitative CCTA and PCAT parameters and myocardial perfusion and flow indices. An economic evaluation was not part of the original study design, and our research team does not have data and sufficient expertise to perform such an analysis at an appropriate level of rigor.
If the reviewer does not mind, we would prefer to keep this aspect beyond the scope of the current publication, as the main focus of our manuscript is on imaging and pathophysiological findings. Nonetheless, we acknowledge that cost-effectiveness considerations represent an interesting direction for future research in this field.
Comments 10: In the Discussion or Limitations section, please include a statement acknowledging potential confounders, particularly the possible effects of medications on MBF and PCAT attenuation.
Response 10: We thank the reviewer for this important comment. We have added a statement to the Limitations section acknowledging this aspect.
The text has been updated in the manuscript on page 16, line 588, and now reads:
[This study has several limitations. First, the relatively small sample size may limit the generalizability of the findings, and further validation in larger cohorts is warranted. Second, despite the use of standardized imaging protocols and post-processing methods, some degree of measurement variability cannot be excluded. To reduce this limitation, we applied standardized threshold values for plaque components as recommended in the latest SCCT consensus documents. This approach increases the comparability of our results with those obtained using dedicated AI-based software. Third, medication use during hospitalization may have influenced both MBF and PCAT attenuation values and thus should be considered potential confounding factors when interpreting the results. Finally, the study did not include an economic assessment of the multimodality imaging approach, as this was beyond the scope and expertise of the research team; however, future studies could explore the cost-effectiveness of such strategies in clinical practice.]
Minor comments
Comments 1: Abstract: move the number of enrolled patients (n = 31) from the Methods section to the Results, as per standard reporting practice.
Response 1: In accordance with the recommendation, we have revised the Abstract by moving the information on the number of enrolled patients (n = 31) from the Methods section to the Results section.
The text has been updated in the manuscript on page 1 in Abstract and now reads:
Methods: Patients with a first episode of AMI were included in the study.
Results: A total of 31 patients (median age 62 [56–70] years) were analyzed, including MICAD (n=21) and MINOCA (n=10).
Comments 2: Abbreviations: Ensure consistency in the use of PCAT, PB, PV, and FAI across text and tables.
Response 2: We thoroughly reviewed the entire manuscript, including tables and figure captions, and standardized the terminology.
Comments 3: Figures 4 and 5: The image captions are detailed but could be shortened; focus on the main imaging findings.
Response 3: We would like to clarify that the figure descriptions include quantitative parameters that are not directly displayed on the images themselves. These details were intentionally included in the captions, as they provide important context for interpreting the clinical examples and comprehensive understanding the imaging findings.
We greatly appreciate and respect the reviewer’s opinion; however, if the reviewer does not object, we would prefer to retain the current version of the figure captions.
Comments 4: References: The list is comprehensive and current. Ensure consistent reference formatting according to Diagnostics journal style.
Response 4: We have reviewed and standardized the reference formatting to ensure full compliance with the Diagnostics journal style guidelines.
- Response to Comments on the Quality of English Language
Point 1: The English is fine and does not require any improvement.
Response 1: We thank the reviewer for the positive feedback regarding the quality of English in our manuscript.
- Additional clarifications
We are especially thankful to the Editors of Diagnostics for their prompt coordination, continuous communication, and efficient handling of the submission and review process.
Additionally, if the editors and reviewers have no objections, we would like to propose a minor revision to the manuscript title. Specifically, we would like to add the word “relationship”, resulting in the following revised title:
“Quantitative Coronary CT Angiography and Pericoronary Adipose Tissue in Acute Myocardial Infarction: Relationship with Dynamic Myocardial Perfusion SPECT”.
The absence of “Relationship” in the initial title of the manuscript was a technical mistake.

Reviewer 2 Report
Comments and Suggestions for Authors
This study investigates the relationship between quantitative CCTA plaque characteristics, PCAT features, and myocardial perfusion parameters in AMI patients. While the research addresses an important clinical question, several methodological and presentation issues need attention.
The study's most critical limitation is the small sample size (n = 31, with only 10 MINOCA patients). This severely limits statistical power and generalizability. The authors should acknowledge this limitation more prominently in the abstract and rather consider this as a pilot/feasibility study rather than definitive findings. I also suggest being more cautious with the conclusions, given the limited sample.
Multiple comparisons were performed without apparent correction (e.g., Bonferroni adjustment). The authors should clarify whether any statistical correction was applied or discuss the risk of type I error.
The exclusion of the left main coronary artery from PCAT analysis requires a stronger justification. A brief paragraph explaining this methodological choice would improve transparency.
There appears to be missing data regarding medication use between ICA and imaging studies, which could influence the results and should be reported. Please address shortly this.
The clinical implications remain insufficiently discussed. How would these findings change patient management? I recommend adding a dedicated “Clinical Perspectives” section outlining potential translational relevance.
Finally, it would be valuable to expand the Discussion section by addressing the potential impact of coronary artery calcium score (CACS) and perivascular fat attenuation index (pFAI) in the context of acute myocardial infarction. A recent review (DOI: 10.3390/jcm13175205) discussed the interplay between these parameters and coronary inflammation, which could provide important context and strengthen the discussion of the role of CCTA.
Author Response
- Summary
We would like to express our sincere gratitude to the reviewer for the careful and thoughtful evaluation of our manuscript, as well as for the constructive comments and valuable insights that have substantially enhanced the clarity, quality, and scientific robustness of our work.
Detailed responses to each point are presented in this document, with all modifications clearly indicated using the comment function in the revised submission for ease of reference.
We deeply appreciate the reviewer’s and editorial team’s time, effort, and guidance throughout the revision process.
- Point-by-point response to Comments and Suggestions for Authors
Comments 1: The study's most critical limitation is the small sample size (n = 31, with only 10 MINOCA patients). This severely limits statistical power and generalizability. The authors should acknowledge this limitation more prominently in the abstract and rather consider this as a pilot/feasibility study rather than definitive findings. I also suggest being more cautious with the conclusions, given the limited sample.
Response 1: We thank the reviewer for this valuable and constructive comment. We have revised the Abstract and the Conclusion. Also, we have revised the first part of Discussion.
The Abstract Conclusion has been updated on page 2, line 52, and now reads:
[Abstract Conclusion:
Patients with MICAD demonstrated a greater extent of atherosclerosis and larger PCAT volume compared with MINOCA. Moreover, PCAT volume demonstrated inverse associations with MBF and CFR, indicating a potential link between PCAT characteristics and microvascular dysfunction].
The first part of Discussion has been updated on page 13, line 436, and now reads:
[Discussion:
1) Although MICAD and MINOCA patients did not differ in age or major risk factors, quantitative CCTA appeared to indicate higher global atherosclerotic burden in MICAD, likely driven by a greater non-calcified plaque component. 2) PCAT volume tended to be higher in MICAD than in MINOCA, whereas PCAT attenuation showed no evident difference between the groups. 3) In the overall cohort, rest perfusion abnormalities as well as rest MBF seemed to be related to CCTA-derived coronary plaque components, whereas CFR did not. 4) PCAT volume showed a weak but statistically significant correlation with stress MBF, but no relationship with semi-quantitatively assessed MP indices.
Overall, our findings suggest a possible trend that quantitative CCTA and PCAT analysis may provide preliminary insights into residual ischemic risk in patients with a first AMI.]
The Conclusion has been updated on page 16, line 609, and now reads:
[Conclusion:
The present study provides preliminary evidence on the relationships between quantitative CCTA-derived plaque and PCAT characteristics and perfusion parameters measured by dynamic SPECT in patients with a first AMI.
Compared with MINOCA, the MICAD phenotype demonstrated a greater atherosclerotic burden, mainly associated with the fibro-fatty plaque component, and a higher PCAT volume, suggesting that the extent of coronary atherosclerosis and PCAT expansion may contribute to the pathophysiological differences between these two conditions. Moreover, PCAT volume showed inverse correlations with MBF and CFR, indicating a potential link between PCAT and microvascular dysfunction.
These findings support the concept that a combined evaluation of coronary PB and PCAT characteristics may provide complementary insights into the pathophysiology of AMI with and without obstructive lesions. The integration of quantitative anatomical (CCTA) and functional (dynamic SPECT) imaging could enhance diagnostic accuracy, deepen understanding of CAD mechanisms, and support more individualized management strategies in this patient population.]
Comments 2: Multiple comparisons were performed without apparent correction (e.g., Bonferroni adjustment). The authors should clarify whether any statistical correction was applied or discuss the risk of type I error.
Response 2: We thank the reviewer for this important comment. We have included a detailed explanation in the Statistical Analysis section, specifying that the Bonferroni correction was applied (Figure 2, Figure 3).
The text has been updated in the manuscript on page 5, line 277, and now reads:
To control for potential type I error due to multiple testing, p-values obtained from the correlation analyses were adjusted using the Bonferroni correction.
Comments 3: The exclusion of the left main coronary artery from PCAT analysis requires a stronger justification. A brief paragraph explaining this methodological choice would improve transparency.
Response 3: We thank the reviewer for this methodological comment. In response, we have added a paragraph to the Methods section explaining this decision.
The text has been updated on page 5, line 220, and now reads:
[In accordance with this validated protocol, PCAT was measured along the proximal 40 mm segments of the right coronary artery (RCA), the left anterior descending artery (LAD) and the left circumflex artery (LCX).
The left main coronary artery (LCA) was excluded from the analysis. This decision was based on the short length of the LCA, its anatomical variability, and close proximity to the aortic wall, which makes it difficult to distinguish pericoronary from paraaortic adipose tissue [9].
Comments 4: There appears to be missing data regarding medication use between ICA and imaging studies, which could influence the results and should be reported. Please address shortly this.
Response 4: We thank the reviewer for this valuable comment. In the revised version of the manuscript, we have clarified this information in the Methods section.
The text has been updated on page 3, line 128, and now reads:
[All patients received standard therapy for AMI according to national guidelines, including dual antiplatelet and lipid-lowering therapy, angiotensin-converting enzyme inhibitors, and beta-blockers in the absence of absolute contra-indications].
As all patients followed the same treatment, we believe this did not introduce additional variability affecting the study outcomes.
Comments 5: The clinical implications remain insufficiently discussed. How would these findings change patient management? I recommend adding a dedicated “Clinical Perspectives” section outlining potential translational relevance.
Response 5: We sincerely thank the reviewer for this insightful and constructive recommendation..
We have added a new subsection entitled “Multimodality Imaging Framework and Clinical Implications” to the Discussion section. The new subsection is on page 15 and now reads:
[Multimodality Imaging Framework and Clinical Implications
The present findings should be viewed in the context of the current non-invasive multimodality imaging framework, which emphasizes the complementary integration of anatomical and functional pathways for the comprehensive assessment of CAD. Rather than being considered as alternative diagnostic strategies, anatomical imaging (e.g., CCTA and quantitative PCAT analysis) and functional imaging (e.g., dynamic SPECT, PET, or stress perfusion cardiovascular magnetic resonance) provide synergistic information that reflects both the structural and physiological components of coronary pathology.
Recent literature has highlighted that this integrated approach enhances diagnostic accuracy, supports individualized patient management, and contributes to a deeper understanding of disease mechanisms across the spectrum of coronary syndromes. As noted by Bergamaschi et al. in their review, combining CCTA-derived plaque and PCAT characteristics with functional parameters (MP, MBF and CFR) represents a pathophysiology-driven paradigm for evaluating CAD [42].
By applying this integrated framework in our cohort of AMI patients, we extend its application beyond chronic coronary syndrome and demonstrate that a combined anatomical-functional strategy may be particularly valuable in the post-infarction setting, where residual flow abnormalities and pericoronary inflammation often coexist. This perspective supports the evolving concept of multimodality imaging as a cornerstone of precision cardiology, enabling more accurate risk stratification and guiding personalized therapeutic decision-making.]
Comments 6: Finally, it would be valuable to expand the Discussion section by addressing the potential impact of coronary artery calcium score (CACS) and perivascular fat attenuation index (pFAI) in the context of acute myocardial infarction. A recent review (DOI: 10.3390/jcm13175205) discussed the interplay between these parameters and coronary inflammation, which could provide important context and strengthen the discussion of the role of CCTA.
Response 6:
We sincerely thank the reviewer for this thoughtful and valuable comment.
In accordance with the reviewer’s suggestion, we have expanded the Discussion section by adding a paragraph addressing the relationship between CACS and pFAI, referencing the recent review (DOI: 10.3390/jcm13175205). However, we should note that CACS was not assessed in our study, as CACS exam was not performed in patients after AMI. Therefore, it was not possible to evaluate the role of CACS and pFAI in the context of AMI within the present study
The text has been updated in the manuscript on page 14, line 510, and now reads:
[Furthermore, evidence suggests that combining FAI with other imaging biomarkers such as the coronary artery calcium score (CACS) may enhance the detection of subclinical inflammation and improve prognostic stratification, as these parameters capture distinct yet complementary aspects of coronary pathology—calcified plaque burden and perivascular inflammatory activity].
- Response to Comments on the Quality of English Language
Point 1: The English is fine and does not require any improvement.
Response 1: We thank the reviewer for the positive feedback regarding the quality of English in our manuscript.
- Additional clarifications
We would like to extend our sincere appreciation to the Editors of Diagnostics for their efficient coordination, timely communication, and attentive management of the submission and peer-review process.
Additionally, if the editors and reviewers have no objections, we would like to propose a minor revision to the manuscript title. Specifically, we would like to add the word “relationship”, resulting in the following revised title:
“Quantitative Coronary CT Angiography and Pericoronary Adipose Tissue in Acute Myocardial Infarction: Relationship with Dynamic Myocardial Perfusion SPECT”.
The absence of “Relationship” in the initial title of the manuscript was a technical mistake

Round 2
Reviewer 1 Report
Comments and Suggestions for Authors
Thank you to the authors for the revisions made, which I believe have enhanced the quality of the final manuscript. I have no further comments.